# Transcriptome-Wide Evaluation Characterization of microRNAs and Assessment of Their Functional Roles as Regulators of Diapause in *Ostrinia furnacalis* Larvae (Lepidoptera: Crambidae)

**DOI:** 10.3390/insects15090702

**Published:** 2024-09-14

**Authors:** Hongyue Ma, Ye Liu, Xun Tian, Yujie Chen, Shujing Gao

**Affiliations:** 1College of Life Sciences and Food Engineering, Inner Mongolia Minzu University, Tongliao 028000, China; ly18247554729@163.com (Y.L.); tianxun@imun.edu.cn (X.T.); chenyujie@imun.edu.cn (Y.C.); 2Institute of Grassland Research of Chinese Academy of Agricultural Sciences, Hohhot 010010, China

**Keywords:** *Ostrinia furnacalis*, diapause, miRNA, regulation, molecular mechanism

## Abstract

**Simple Summary:**

Diapause is a state in which insects respond to environmental changes, leading to developmental stagnation, which is crucial in the life history of insects. miRNA regulates the expression of genes after transcription and participates in the regulation of important biological processes of insect growth and development. In this study, we screened differentially expressed miRNAs (DEMs) in non-diapause/diapause and diapause/non-diapause comparison groups of *Ostrinia furnacalis* and predicted their target genes. The expression patterns of key target genes Kr-h, JHE, JHEH, FOXO, Cry, and Per in diapause-related metabolic pathways at different stages of diapause were opposite to those of miRNAs, suggesting their regulatory roles in the diapause process. This study improves the scientific understanding of diapause in *O. furnacalis*; the learning can be applied to other insects.

**Abstract:**

microRNAs (miRNAs) function as vital regulators of diapause in insects through their ability to post-transcriptionally suppress target gene expression. In this study, the miRNA of *Ostrinia furnacalis,* an economically important global crop pest species, was characterized. For the included analyses, 9 small RNA libraries were constructed using *O. furnacalis* larvae in different diapause states (non-diapause, ND; diapause, D; diapause-termination, DT). The results identified 583 total miRNAs, of which 256 had previously been identified, whereas 327 were novel. Furthermore, comparison analysis revealed that 119 and 27 miRNAs were differentially expressed in the D vs. ND and DT vs. D, respectively. Moreover, the expression patterns of their miRNAs were also analyzed. GO and KEGG analysis of the target genes of differentially expressed miRNAs highlighted the importance of these miRNAs as diapause regulators in *O. furnacalis*, especially through metabolic processes, endocrine processes, 20-hydroxyecdysone, and circadian clock signaling pathways. In summary, this study highlighted the involvement of specific miRNAs in the control of diapause in *O. furnacalis*. To the best of our knowledge, this is the first study to identify miRNA expression patterns in *O. furnacalis*, thereby providing reference and novel evidence enhancing our current understanding of how small RNAs influence insect diapause.

## 1. Introduction

miRNAs are ~22-nucleotide (nt) single-stranded RNAs that lack coding potential; however, they can suppress target gene expression via the post-transcriptional inhibition of translation or the induction of complementary mRNA degradation [1]. miRNAs expression has been comprehensively studied across species of insects, which has indicated that specific miRNAs play an important role in the control of physiological and biological pathways [2]. In the previous literature, high-throughput sequencing strategies have been employed for the identification of miRNAs in insect species, including *Bombyx mori* [3], *Helicoverpa armigera* [4], and *Plutella xylostella* [5], and provided insights into tissue- and stage-specific patterns of miRNA expression. Furthermore, a study indicated that in *Tribolium castaneum*, RNAi knockdown specific miRNAs (miR-6-3p, miR-9a-3p, miR-9d-3p, iR-11-3p, and miR-3p) can impair wing development and promote other deformities [6]. Therefore, comprehensive studies on differential miRNA expression patterns are essential for a better understanding of how these noncoding RNAs modulate growth and development in insects.

Diapause is a state of developmental arrest activated by exposure to aversive environmental conditions [7]. Reynolds et al. [8] compared diapause and non-diapause pupae of the *Sarcophaga bullata* and indicated the differential expression of 10 conserved miRNAs. A total of seven differentially expressed miRNAs were reported in *Aedes albopictus* during diapause, and the target genes of these miRNAs were found to be involved in diapause-related immune response, lipid metabolism, development, and ecdysteroids pathways [9]. These findings suggest that miRNAs are essentially involved in the control of insect diapause. Diapause is modulated by the control of hormone production and gene expression, with miRNAs, given their ability to shape patterns of gene expression, being prime candidate mediators of this process. Several studies have indicated that the levels of certain miRNAs alter during diapause, such as miR-2765-3p in *Galeruca daurica,* which was demonstrated to target FoxO to regulate reproductive diapause [10]. However, the specific processes through which altered miRNA expression affects the overall regulatory network of diapause initiation, maintenance, and termination remain elusive.

*Ostrinia furnacalis* is an important pest species in the family Crambidae (Lepidoptera) that is often referred to as the Asian corn borer or stem borer. It has been observed that *O. furnacalis* has spread to East Asia, specifically in China. The highly adaptable nature of this species allows it to cause significant damage to crops, including corn. It is reported that *O. furnacalis* usually reduces maize production by about 10% and can reduce production by more than 30% when it is severe, incurring major agricultural losses such that it is classified by the Chinese Ministry of Agriculture as a national first-class crop pest. *O. furnacalis* shows complete metamorphosis that goes through four stages: egg, larva, pupa, and adult, among which larva is the most harmful. Furthermore, *O. furnacalis* is a facultative diapause insect species, which overwinters as old larvae. Therefore, *O. furnacalis* larvae are ideal materials for the study of diapause in lepidoptera insects, and its diapause process is determined by both environmental factors and genetic genes. However, the molecular mechanisms that govern diapause in this species remain undetermined. It has been found that there are numerous horizontally transferred genes that regulate reproduction and metamorphosis in lepidopteran pests [11]. Studies have found that circadian rhythm-related pathways are important in the regulation of diapause in this pest species, culminating in the successful identification of a master regulatory site associated with the control of diapause [12]. Changes in the genes found at this site can contribute to altered photoperiod sensitivity, thus modulating diapause incidence and termination. However, there are no studies on the molecular regulation of diapause in *O. furnacalis*. Therefore, exploring the molecular mechanism and regulatory network of diapause in *O. furnacalis* larvae is not only a major scientific issue that needs to be addressed urgently but also has important economic significance.

## 2. Materials and Methods

### 2.1. Insect Materials

*O. furnacalis* specimens were fed an artificial diet and maintained at the Inner Mongolia Minzu University of College of Life Sciences and Food Engineering. When larvae reached the 5th instar stage (1.2–3 mm), samples were collected in the non-diapause [ND; 26 °C, 16L:8D (16 h light/8 h dark cycle, 80% ± 1 relative humidity (RH))] and diapause (D; 22 °C, 8L:16D, 80% ± 1 RH) states. In addition, 5th instar larvae in the diapause state were placed at 4 °C for 15 days, followed by transfer into an incubator (26 °C, 16L:8D, 80% ± 1 RH), allowing for the collection of diapause-terminated (DT) samples following continued feeding of these larvae. Each of these treatment groups had 3 replicate samples, which were ground in liquid nitrogen and stored at −80 °C until the sRNA-seq and qPCR analyses.

### 2.2. Deep Sequencing

TRIzol (Thermofisher, 15596018, Waltham, MA, USA) was used to extract the total RNA from each sample, as per the kit’s instructions. The quantity and purity of extracted RNA were assessed using a NanoDrop ND-1000 spectrophotometer (NanoDrop, Wilmington, DE, USA). Furthermore, RNA integrity was measured with an Agilent 2100 Bioanalyzer (Agilent, Santa Clara, CA, USA). Those samples exhibiting a 28S/18S > 0.7 and an RIN > 7.0 were used to prepare small RNA (sRNA) libraries with the TruSeq Small RNA Sample Prep Kits (Illumina, San Diego, CA, USA). After isolation, total RNA samples were subjected to size fractionation, with those sRNAs 10–30 nucleotides in length then being isolated via gel separation, purified, ligated with 3′ and 5′ adapters, reverse transcribed, and enriched via PCR. Purified fragments were then sequenced (single-ended 50 bp; SE50) with an Illumina Hiseq 2500 instrument (LC BioTechnology Co., Ltd., Hangzhou, China).

### 2.3. Sequencing Data Processing

Raw reads were processed using the ACGT101-miR v4.2 package (LC Sciences, Houston, TX, USA) to remove repeats, adapters, low-complexity sequences, junk sequences, and common RNA types (tRNAs, rRNAs, snRNAs, snoRNAs). The remaining unique 18–26 nt long reads were then mapped to miRBase 22.1 [12] to identify known miRNAs. Furthermore, BLAST was used to identify novel 3p- and 5p-derived miRNAs [13]. For sequence alignment, variations at the 5′ and 3′ ends and a maximum of one internal sequence mismatch were permitted. The BOWTIE was employed to compare unmapped sequences with the *O. furnacalis* genome (Accession number: PRJNA429307), allowing a maximum of one mismatch [14]. Sequences containing hairpin RNA structures were predicted based on the flanking 80 nt sequences with the RNA fold program (http://rna.tbi.univie.ac.at/cgi-bin/RNAWebSuite/RNAfold.cgi, accessed on 16 May 2023) [15]. The flanking sequences for mapped reads then underwent secondary structural analyses aimed at the prediction of pre-miRNA sequences [16]. Sequences that formed stem-loop structures with flanking sequences and that were present in stem regions were regarded as *O. furnacalis* miRNA candidates. Modified global normalization was performed to control the copy number differences between the samples [17].

### 2.4. Differentially Expressed miRNA Analyses

Differentially expressed miRNAs (DEMs) were identified via Student’s *t*-tests based on the normalized deep sequencing counts. When diapause states (D/ND and DT/D) were compared, the acquired DEMs were designated as miRNAs at an adjusted *p* < 0.05 and |log2 (FC)| > 1.

### 2.5. Target Prediction and Functional Annotation

To identify target genes associated with the most abundant miRNAs, the alignment of miRNA sequences to the NCBI *O. furnacalis* genome database (Accession number: PRJNA429307) was performed via TargetScan v5.0 [18] and Miranda v3.3a [19] to identify possible sites of miRNA binding. Then, the targets overlapping between both of these algorithms were identified as the final set of theoretical miRNA targets (target gene thresholds: TargetScan score ≥ 50, Miranda energy < −10). These putative targets were then mapped to the GO database (http://www.geneontology.org/, accessed on 19 May 2023), and gene numbers associated with each GO term were assessed using Blast2GO [20]. Consistent with previous studies [21], the KOBAS program [22] was employed for KEGG pathway analyses (http://www.genome.jp/kegg/, accessed on 19 May 2023). Significantly enriched GO and KEGG pathways (*p* < 0.05) were identified via Fisher’s exact test.

### 2.6. qPCR Validation

To confirm the accuracy of the sRNA-seq results, the RNA samples that were used for sRNA library preparation were subjected to qPCR analyses. Briefly, miRNA 1st Strand cDNA Synthesis Kit (by tailing A) (Vazyme, Nanjing, China) was used to prepare cDNA, which was then used for qPCR amplification using an ABI-7500 instrument (Applied Biosystems, Waltham, MA, USA) and the Taq pro-Universal qPCR Master Mix (2×) (Vazyme). The reaction parameters were as follows: 7 min at 95 °C; 40 cycles of 95 °C for 20 s and 60 °C for 90 s; melt curve analysis from 60 to 95 °C. The experiment was repeated with three biological replicates (each replicate included 5 insects) and 4 or more technical replicates per biological replicate. U6 snRNA was employed as a normalization control for miRNA analysis. The relative expression was assessed using the 2^−∆∆Ct^ method [23]. The primers used in this analysis were designed with Primer-BLAST (http://www.ncbi.nlm.nih.gov/tools/primer-blast/, accessed on 23 March 2023) (Appendix A).

## 3. Results

### 3.1. Analyses of sRNA Data

The sequencing analyses yielded 88.12 million raw reads that have been uploaded to the NCBI Sequence Read Archive (PRJNA1018261). After low-quality reads, adapter sequences, and RNAs < 18 nucleotides or > 26 nucleotides long (ACGT101-miR) had been removed, the dataset consisted of ~64.61 million clean reads were found, including 22,058,844 (34.14%) from the ND group, 20,360,757 (31.52%) from the D group, and 22,181,952 (34.34%) from the DT group (Appendix A). In terms of their length distributions, these total sRNAs exhibited a bimodal pattern with peaks at 20 and 22 nt, consistent with the typical length of miRNAs (Appendix A). After aligning and annotating, these sRNA reads were categorized into 9 groups. Approximately 31.8 million reads that were associated with known and novel miRNAs were retained for analysis, including 4,635,549 and 5,282,122 respective reads for the ND group, 5,009,956 and 5,960,892 for the D group, and 5,522,041 and 5,459,801 for the DT group (Table 1).

### 3.2. miRNA Expression Profiles

From the three analyzed libraries, 265 conserved and 327 novel miRNAs were identified, respectively (Appendix A), with a total of 312, 308, and 322 miRNAs in ND, D, and DT states samples, respectively (Appendix A). Hierarchical clustering analyses of the expression profiles for these miRNAs revealed particular sets of state-specific miRNA expression patterns (Figure 1).

### 3.3. DEMs Identification

When comparing the D and ND groups (D/ND), 119 significant DEMs were identified (62 upregulated, 57 downregulated) (Figure 2, Appendix A), with the top three most upregulated miRNAs being miR-14, miR-277, and miR-2766-3p, whereas the three most downregulated miRNAs were miR-10-5p, miR-305-5p, and miR-3344-p5 (Table 2). For the comparison of the DT and D groups (DT/D), 27 DEMs were identified (14 upregulated, 13 downregulated) (Figure 2, Appendix A), among which the three most significantly upregulated were miR-6307, miR-970-5p, and PC-3p-81145, whereas the three most downregulated were PC-3p-431, PC-5p-66538, and PC-3p-230 (Table 2). When these two comparisons were evaluated for overlap (D/ND and TD/D), 134 total DEMs were identified, of which 12 DEMs co-existed in both comparisons (Table 2), suggesting that they may crucially regulate diapause in *O. furnacalis* larvae.

### 3.4. Predictive Identification and Functional Annotation of DEM Targets

The 134 identified DEMs were aligned with the *O. furnacalis* database (GenBank accession number: PRJNA429307), which revealed 72,403 putative target transcripts that were then subjected to GO and KEGG annotation analyses. For the D/ND comparison, the predicted targets of 119 DEMs were significantly enriched in 363 GO terms (Appendix A), the top 20 of which are presented in Figure 3A. The most strongly enriched biological process (BP) terms included ‘positive regulation of transcription by RNA polymerase II’, ‘negative regulation of transcription’, and ‘compound eye development’. Moreover, the most enriched cellular component (CC) terms included ‘transcription regulator complex’, ‘plasma membrane’, and ‘cleavage furrow’, while the most enriched molecular function (MF) terms were ‘DNA-binding transcription factor activity’, ‘protein binding’, and ‘transcription factor binding’. The KEGG analyses revealed 24 significantly enriched pathways across five classification groups (Figure 4A, Appendix A). The enriched metabolism-related pathways included the ‘glycerophospholipid metabolism’, ‘glycosaminoglycan biosynthesis’, and ‘ether lipid metabolism’ pathways. In addition, the enriched pathways related to genetic information processing included ‘SNARE interactions in vesicular transport’. Additionally, the enriched environmental information processing-related pathways included ‘the MAPK signaling pathway-fly’, ‘FOXO signaling pathway’, and ‘TGF-beta signaling pathway’. Furthermore, the enriched pathways associated with the cellular process included ‘endocytosis’, ‘regulation of actin cytoskeleton’, and ‘tight junction’, whereas the enriched organismal system-related pathways included the ‘dorsoventral axis formation’, ‘Toll and Imd signaling’, and ‘longevity regulating pathway-multiple species’ pathways.

For the DT/D comparison, the target genes of 27 DEMs were significantly enriched in 397 GO terms (Appendix A, Figure 3B). The most significantly enriched BP terms included ‘axon guidance’, ‘dorsal closure’, and ‘compound eye development’, whereas the most enriched CC terms included ‘plasma membrane’, ‘cleavage furrow’, and ‘transcription regulator complex’, while the most enriched MF terms were ‘protein binding’, ‘RNA polymerase II transcription regulatory region sequence-specific DNA binding’, and ‘DNA-binding transcription factor activity’. Furthermore, the KEGG analyses revealed that these genes were enriched in 19 pathways in the five classifications (Figure 4B, Appendix A). Enriched metabolism-related pathways included ‘glycerophospholipid metabolism’, ‘glycosaminoglycan biosynthesis’, and ‘pantothenate and CoA biosynthesis’ pathways. Moreover, the identified enriched genetic information processing-related terms included the ‘SNARE interactions in vesicular transport’ and ‘protein processing in endoplasmic reticulum’ pathways, whereas the enriched environmental information processing pathways included ‘MAPK signaling pathway-fly’, ‘Wnt signaling pathway’, and ‘Hippo signaling pathway-multiple species’, cellular process-related pathways included ‘tight junction’, ‘mitophagy-animal’ and ‘endocytosis’, and organismal system-related pathways included the ‘dorsoventral axis formation’, ‘circadian rhythm-fly’, and ‘Toll and Imd signaling’ pathways.

Moreover, 20-hydroxyecdysone (20E) plays an important role in regulating insect diapause. The 20E signaling pathway was also significantly enriched for the genes targeted by 13 DEMs, including PC-5p-10843_308, PC-5p-39767_82, PC-5p-44721_69, miR-305-5p, miR-10-3p, miR-34-5p, miR-92b, and miR-8-5p targeting ecdysone receptor (EcR), and PC-5p-10843_308, PC-5p-378_12754, miR-190-5p, miR-33-5p, and let-7a targeting Ultraspiracle (USP).

These functional enrichment analysis results underscore the potentially vital roles that the identified DEMs may play in the control of *O. furnacalis* diapause through their post-transcriptional control of specific target genes.

### 3.5. qPCR Validation of sRNA-Seq Data

To validate the sRNA-seq results, eight identified DEMs were randomly selected and analyzed via qPCR (Figure 5). The data revealed similar expression patterns of these eight miRNAs as compared to the sRNA-seq dataset, confirming the reliability of initial analyses.

### 3.6. Expression Patterns of DEMs and Their Target Genes

To analyze the molecular mechanism of miRNA regulating the expression level of its target genes and thus affecting the diapause of *O. furnacalis*, the expression profiles of DEMs and their candidate target genes were analyzed. The identified candidate target genes were found to be involved in the regulation of the key metabolic pathways of insect juvenile hormones, ecdysone, and circadian clock signals. The qPCR was performed at different diapause stages to identify the relative expression profiles of the selected six signaling-related target genes, including *Krüppel-homolog 1* (*Kr-h1*), *juvenile hormone esterase* (*JHE*), *juvenile hormone epoxide hydrolase* (*JHEH*), *forkhead box protein O* (*FOXO*), *cryptochrome* (*Cry*), and *period circadian protein* (*Per*). The results showed that the expression pattern of miRNA and the target gene had an opposite trend, confirming that there may be a potential target relationship (Figure 6). However, the specific target mechanism needs further experimental verification and will be focused on in our future work.

## 4. Discussion

In this study, sRNA libraries corresponding to three diapause states of *O. furnacalis* were established to identify expression changes in regulatory miRNAs, highlighting the abundance of sRNAs in this insect species. These sRNAs fit a bimodal length distribution consistent with what has been described in *Bombyx mori* [24], *Nilaparvata lugens* [25], and *Bactrocera dorsalis* [26]. These results are not universal, however, with a peak sRNA size of 22 nt having been reported for species including *Locusta migratoria* [27], *Culex quinquefasciatus* [28], and *Plodia interpunctella* [29].

This study identified a large number of DEMs in *O. furnicalis* with varying degrees of up- or downregulation in diapause states, supporting their potentially complex regulatory roles in diapause. Furthermore, certain miRNAs were found to be exclusively expressed in a single diapause state, which may indicate that they play diapause phase-specific roles. In a prior comparison of *Aedes albopictus* larvae, researchers identified 7 DEMs between the diapause and non-diapause states (miR-1-3p, miR-14-5p, miR-183-5p, miR-282-5p, miR-286b-3p, miR-3942-3p, bantam-5p), suggesting their potential involvement in mediating larval diapause [9]. Here, many DEMs were identified across three diapause states, in particular, miR-970-5p, miR-306a-5p, PC-3p-230, and miR-277, indicating that a complex post-transcriptional regulatory network may modulate diapause in *O. furnacalis*.

Insulin, JH, and ecdysone are three major hormones that regulate reproduction and diapause in insects [30]. Studies have shown that JH agonists repress the expression of *miR*-*927* in *Drosophila melanogaster* [31], and 20E inhibits the expression of *miR*-*281* in *B. mori* [32]. This indicates that hormone treatment can regulate the expression patterns of miRNA and target genes and has a potential effect on the occurrence of diapause. The literature has indicated that miR-14 was a negative regulator of insulin signaling and a promoter of *Drosophila* nutrient storage. Here, significant miR-14-5p upregulation was observed during diapause in *O. furnacalis*, indicating that it may regulate nutrient storage and metabolism during diapause. Several studies have validated the importance of miRNAs in the control of appropriate target gene expression levels [33]. Therefore, the identification of miRNA target genes is important for understanding the biological functions of these noncoding RNAs. Here, predictive analyses identified many putative miRNA target genes throughout the *O. furnacalis* genome. Although the relationship between these miRNAs and their target genes needs further verification, this study provides insights into the potential processes of specific miRNAs regulating the diapause of *O. furnacalis* larvae.

In insects, the steroid hormone 20-hydroxyecdysone (20E) is synthesized in and secreted from the prethymus, whereupon it signals through a heterodimeric receptor consisting of Ultraspiracle (USP) and ecdysone receptor (EcR) to activate 20E early induction genes, thereby influencing larval and pupal diapause-related processes [34]. In the absence of 20E, larval diapause is induced. In this study, 13 DEMs targeting the transcripts encoding USP and EcR were identified. A previous study indicated that in *B. mori*, 20E treatment inhibits virus-induced enhanced locomotory activity, whereas let-7 regulates larval molting and metamorphosis by targeting genes involved in the ecdysone pathway [35,36]. In this study, let-7a was significantly downregulated during diapause (D/ND). In studies of *Tribolium castaneum*, overexpression of miR-8 [37], miR-8-3p [38], and miR-34 [39] have all been reported to impact the process of ecdysteroid biosynthesis, delay development, and/or induce morphological defects in embryos, eyes, legs, and wings, and promote endocrine defects during the pupation, eclosion, and wing developmental processes [6]. Furthermore, this study identified that miR-8-5p, miR-34-5p, and miR-34-3p were differentially expressed in *O. furnacalis* larvae during the diapause and diapause-termination states, indicating that these miRNAs may coordinate the endocrine and developmental pathways involved in diapause maintenance in this insect species.

Diapause induction is markedly associated with the circadian clock signaling pathway. The most extensively studied genes that regulate circadian rhythmicity include Timeless (Tim), Cycle (Cyc), Clock (Clk), Period (Per), and Cryptochrome (Cry) [40]. A study on *Riptortus stris* revealed that the RNAi-based silencing of *Cry-m* was associated with the entry of control insects into diapause under short-day conditions, whereas *Cry*-*m* RNAi treatment was associated with a significant increase in the rate of individual development [41]. In addition, Kotwica-Rolinska et al. [42] reported that in *Pyrrhocoris apterus*, both Cry-m and pigment-diffusing factor (PDF) essentially mediate diapause induction. *PDF* knockout in *Culex pipens* has been observed to promote diapause even under a long photoperiod [43]. A total of 36 DEMs were identified that are associated with four target genes related to circadian clock signaling pathways. Appendix A suggests a central role for these miRNAs in the regulation of diapause induction in *O. furnacalis*. Next, we will further investigate the specific functions of these target genes in diapause regulation, thereby gaining a deeper understanding of miRNA regulatory networks. This will help us to better understand the molecular mechanism of diapause regulation in *O. furnacalis* and may provide potential management strategies for agricultural production.

## 5. Conclusions

In summary, this study performed the sRNA-seq analyses of *O. furnacalis* larvae and identified 265 and 327 known and novel miRNAs, respectively, with significant expression differences across diapause stages. Furthermore, the Target prediction analyses revealed the potential involvement of these miRNAs in the regulation of various crucial physiological and biological processes associated with different diapause stages in *O. furnacalis* (Figure 7). This study provides a valuable new resource for further studies focused on the roles of small RNAs in the control of insect diapause and offers a foundation for future mechanistic research focused on the regulatory network governing the diapause process in *O. furnacalis*.

## Figures and Tables

**Figure 1 insects-15-00702-f001:**
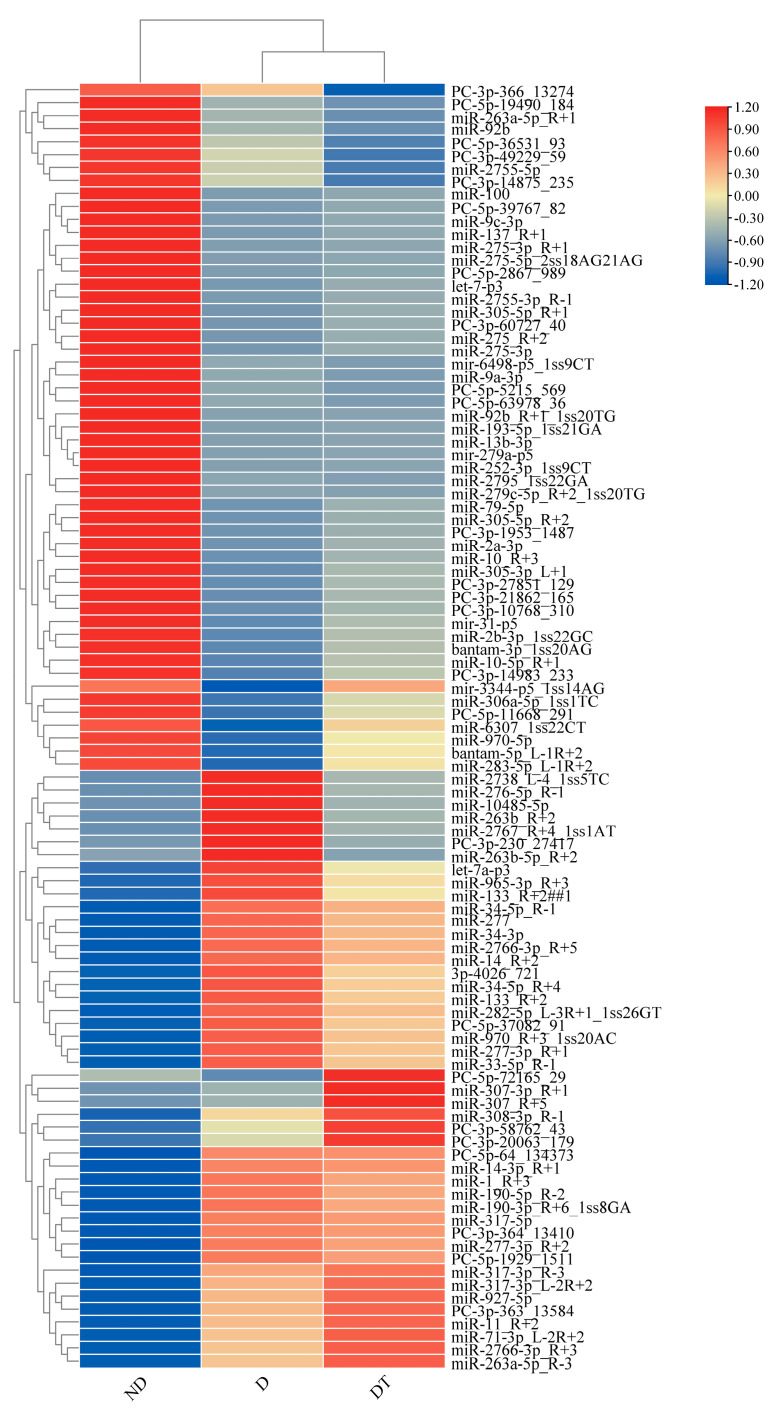
Clustering analysis of miRNAs associated with different diapause states in *O. furnacalis*. Data are presented as the average normalized values of three biological replicates. ND: Non-diapause; D: diapause; DT: diapause-termination.

**Figure 2 insects-15-00702-f002:**
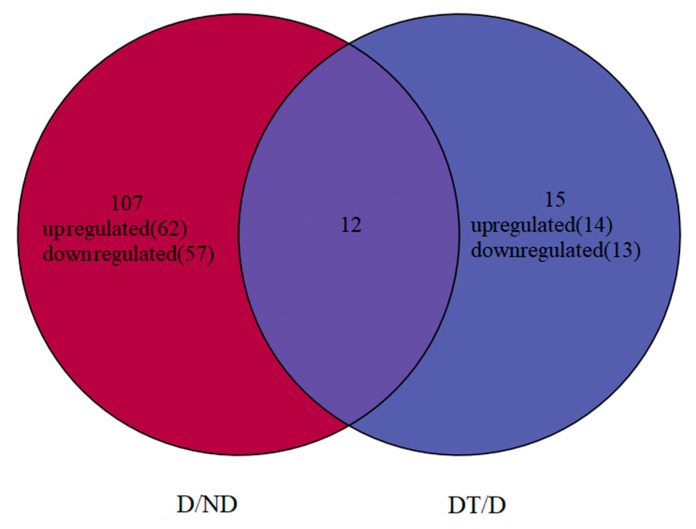
Venn diagram analysis of differentially expressed miRNAs between the compared D/ND and DT/D diapause states. ND: Non-diapause; D: diapause; DT: diapause termination.

**Figure 3 insects-15-00702-f003:**
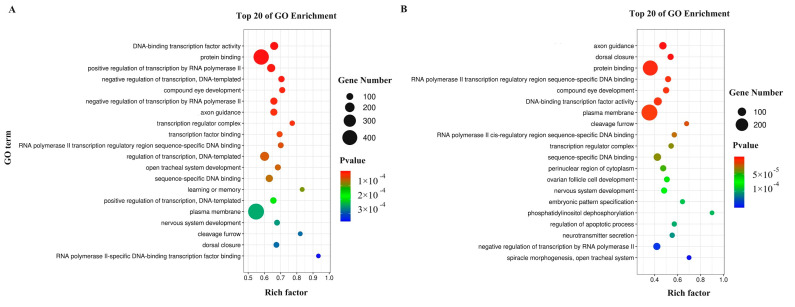
The top 20 enriched GO terms are associated with DEM target genes in *O. furnacalis*. (**A**) D/ND; (**B**) DT/D.

**Figure 4 insects-15-00702-f004:**
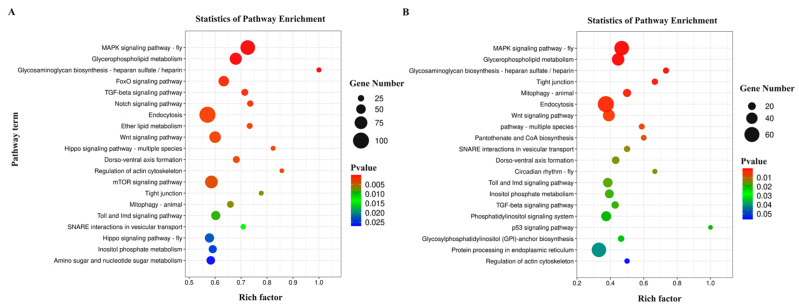
KEGG enrichment analyses of DEM target genes in *O. furnacalis*. (**A**) D/ND; (**B**) DT/D.

**Figure 5 insects-15-00702-f005:**
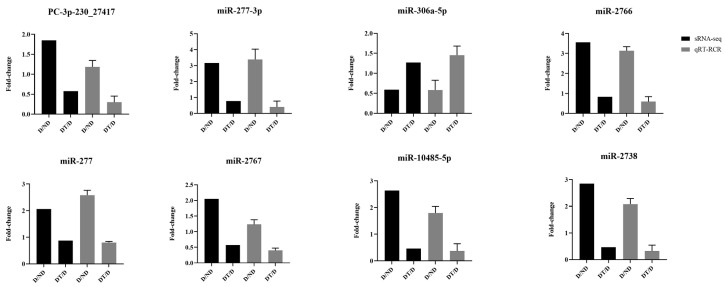
qPCR validation of sRNA-seq results.

**Figure 6 insects-15-00702-f006:**
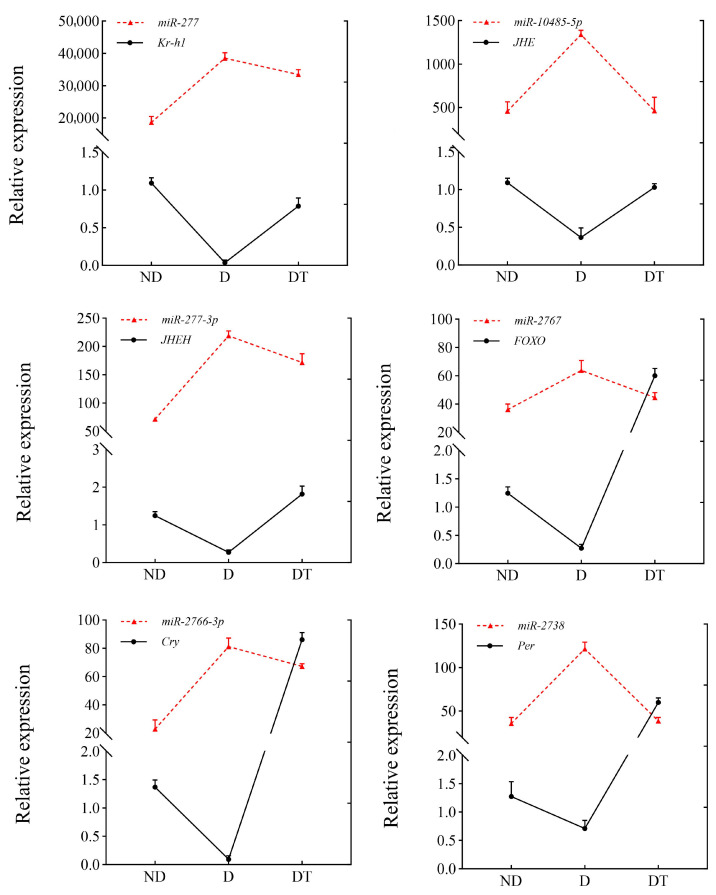
Relative expression profiles of miRNA and target genes in the ND, D, and DT states of *O. furnacalis*. Relative expression levels of miRNA and target genes were normalized to U6 and *β*-actin, respectively. Each point represents the mean relative expression level, and the error bars indicate standard error (SE).

**Figure 7 insects-15-00702-f007:**
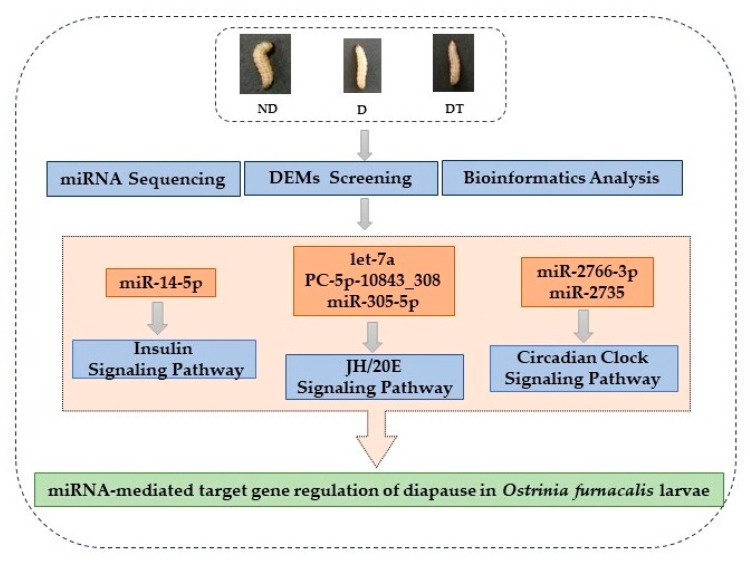
The regulatory model of miRNA in the diapause of *O. furnacalis*.

**Table 1 insects-15-00702-t001:** The composition of small RNA classes in three diapause states of *O. furnacalis*.

Type	ND		D		DT		Combined	
	Reads	%	Reads	%	Reads	%	Reads	%
Known miRNA	4,635,549	20.24	5,009,956	23.91	5,522,041	24.28	15,167,546	22.77
Novel miRNA	5,282,122	23.07	5,960,892	28.45	5,459,801	24.00	16,702,815	25.08
rRNA	638,409	2.79	507,609	2.42	439,204	1.93	1,585,222	2.38
tRNA	303,522	1.33	231,957	1.11	212,673	0.93	748,152	1.12
snoRNA	2560	0.01	3142	0.01	3076	0.01	8778	0.01
snRNA	11,402	0.05	7285	0.03	16,048	0.0	34,735	0.05
other Rfam RNA	122,015	0.53	95,717	0.46	83,110	0.37	300,842	0.45
Transcriptome (mRNA)	3,137,096	13.70	2,377,816	11.35	2,713,959	11.93	8,228,871	12.36
Unannotated	8,767,533	38.29	6,758,616	32.26	8,295,952	36.47	23,822,101	35.77
Total (clean reads)	22,900,208	100	20,952,990	100	22,745,864	100	66,599,060	100

rRNA: ribosomal RNA; tRNA: transfer RNA; snoRNA: small nucleolar RNA; snRNA: small nuclear RNA; Transcriptome (mRNA): total mRNA; Unannotated: small RNAs not annotated.

**Table 2 insects-15-00702-t002:** DEMs associated with different diapause states in *O. furnacalis* larvae.

**miRNAs**	**Log2 (Fold Change)**	***p*-Value**	**Up/Down**
D/ND			
miR-14	2.54	9.93 × 10^−5^	Up
miR-277	1.04	1.70 × 10^−4^	Up
miR-2766-3p	1.83	3.39 × 10^−4^	Up
miR-10-5p	−1.43	7.87 × 10^−5^	Down
miR-305-5p	−3.43	6.90 × 10^−4^	Down
mir-3344-p5	-inf	9.10 × 10^−4^	Down
DT/D			
miR-6307	0.50	9.02 × 10^−3^	Up
miR-970-5p	-inf	1.13 × 10^−2^	Up
PC-3p-81145	-inf	1.22 × 10^−2^	Up
PC-3p-431	−0.41	2.27 × 10^−3^	Down
PC-5p-66538	-inf	8.90 × 10^−3^	Down
PC-3p-230	−0.78	1.12 × 10^−2^	Down
**Common DEMs Overlapping between the Two Comparisons**			
miRNAs	D/ND		DT/D
miR-970-5p	Down		Up
miR-306a-5p	Down		Up
mir-3344-p5	Down		Up
PC-3p-230	Up		Down
miR-277	Up		Down
miR-263b	Up		Down
miR-263b-5p	Up		Down
miR-2738	Up		Down
miR-2767	Up		Down
miR-277-3p	Up		Down
miR-10485-5p	Up		Down
miR-2766-3p	Up		Down

## Data Availability

Data are contained in the article and Appendix A. The microRNA data have been uploaded to NCBI (GenBank accession number: PRJNA1018261).

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
