# Peer review of "Transcriptome-Wide Evaluation Characterization of microRNAs and Assessment of Their Functional Roles as Regulators of Diapause in Ostrinia furnacalis Larvae (Lepidoptera: Crambidae)"

_insects, 2024, doi:10.3390/insects15090702_

Round 1

Reviewer 1 Report

Comments and Suggestions for Authors

This article is very interesting as it explores the significant role of specific miRNAs in the regulation of diapause in Ostrinia furnacalis, providing new insights into the molecular mechanisms of insect diapause. The identification of the relationship between miRNAs and their target genes, particularly the expression patterns of key diapause-related target genes such as Kr-h, JHE, JHEH, FOXO, Cry, and Per, reveals that these genes exhibit expression patterns that are correlated with those of the miRNAs. This suggests their regulatory roles in the diapause process. The results are generally interesting; however, I have several concerns that I would like to discuss further.

  1. In the introduction section, I recommend elaborating on the importance of studying Ostrinia furnacalis (an important agricultural pest), especially since this species is not a model organism and may not directly attract reader interest. It would be better to explain why Ostrinia furnacalis is significant, highlighting its characteristics as a Lepidopteran insect, and explain why you study. This could be approached in various ways, such as its impact on agriculture, its specific or unique traits, or the presence of numerous horizontally transferred genes in Lepidopteran pests (see https://doi.org/10.1016/j.cell.2022.06.014).
  2. Regarding figure arrangement, I suggest organizing the figures into PDF or SVG vector graphics. From Figures 1 and 2, it is shown that the x,y-axis axes are not clearly labeled, making it difficult to understand. Specifically, Fig. 1 presents technique control data, which is not a critical result and could be placed in a supplemental figure. Fig. 2 does not convey the results, and the overall color scheme is not cohesive. Additionally, the Venn diagram in Fig. 3 uses only three colors, which does not necessitate color differentiation. This feedback applies to the following figures as well.
  3. Although the results include qPCR validation of sRNA-seq as experimental verification, I would like to know if the identification of functional miRNAs implies that knocking out or knocking down these miRNAs would directly affect the expression patterns of the key diapause-related target genes (such as Kr-h, JHE, JHEH, FOXO, Cry, and Per), and subsequently influence the diapause of this insect.
Comments on the Quality of English Language

Minor comments:

L330, "Here, a total of 36 DEMs were identified by 4 target genes related to circadian clock signaling pathways”, suggest to be revised to "Here, a total of 36 DEMs were identified that are associated with 4 target genes related to circadian clock signaling pathways."

L333, "Next, we will furtherly study the specific functions of these target genes" suggests to be revised to "Next, we will further investigate the specific functions of these target genes in diapause regulation."

L343, "This study provides a valuable new resource for further studies focused on the roles that small RNAs in the control of insect diapause” suggests to be revised to "This study provides a valuable new resource for further studies focused on the roles of small RNAs in the control of insect diapause."

L30, "the expression patterns of these differentially expressed miRNAs were also analyzed" is redundant. It could be simplified to "the expression patterns of these miRNAs were analyzed."

L350, "Table S3: Identification and norm values of novel mi- 350 croRNAs (miRNAs) from the three diapause states of Ostrinia furnacalis" suggest to be revised to "Table S3: Identification and norm values of novel microRNAs (miRNAs) from the three diapause states of Ostrinia furnacalis."

The MS uses both "miRNAs" and "microRNAs" interchangeably. It is advisable to choose one term and use it consistently throughout the manuscript to avoid confusion.

Reviewer 2 Report

Comments and Suggestions for Authors

The manuscript with characteristics of diapause-related micro RNAs in a lepidopteran pest Ostrinia furnacalis. The study is scientifically sound and, although preliminary (which the authors don’t deny), is important for to form the basis and formulate hypotheses for future research in this field. The plan of the paper is clear and quite easy to follow.

I have a few concerns and suggestions (see below).

All figures: please increase figure quality. They look like they have been heavily donwsampled, and text is hardly readable.

Figure 3 and the following ones, as well as the corresponding text: I do not fully understand why D/ND and DT/D were compared exactly in this direction. As for me, it would make sense to compare ND/D and DT/D as (1) these will have a common reference (control) condition and (2) both will relate a non-diapause state to a diapause state. Probably the logic was different (please explain).

L116 “O. furnacalis genome (Accession number: PRJNA429307),”: the BioProject is from 2019. Is the assembly still unpublished?

As far as I can see, this manuscript at least mentiones the assembly:

https://www.sciencedirect.com/science/article/abs/pii/S0965174821001211

Please consider

L118 “(http://rna.tbi.univie.ac.at/cgi-bin/RNAfold.cgi).”:

- is would also be great to cite an appropriate manuscript to honour the authors.

- the link did not check I suppose : http://rna.tbi.univie.ac.at/cgi-bin/RNAWebSuite/RNAfold.cgi

L125 “Differentially expressed miRNAs (DEMs) were identified via Student’s t-tests based

on the normalized deep sequencing counts”: was correction for multiple comparisons applied?

L151: how was relative expression compared between groups?

L202: “which revealed 72,403 putative target transcripts” vs L258 “The identified candidate target genes”: how were these genes selected?

L276 “64.61 clean reads”: “64.61 M clean reads” ?

Comments on the Quality of English Language

There are a few typos or small grammatical errors, but I’m quite sure a standard proofreading procedure is enough to solve the issue. Please find some examples below, but the list is by no means exhaustive.

L. 18-19 “As the key target genes of diapause relevant

metabolic pathways, the expression patterns” : “The expression patterns of the key target genes of diapause relevant metabolic pathways” ?

L75 “overwinters as mature and old larvae”: “overwinters as either imago or old larvae”?

Round 2

Reviewer 1 Report

Comments and Suggestions for Authors

I appreciate the effort of the author in resolving most of the concerns I raised. However, I would like to further discuss Comment 3, as this aspect is central to your study, which explores the relationship between microRNAs and insect diapause.

1. Although the results include qPCR validation of sRNA-seq as experimental verification, my main concern is still whether the identification of functional miRNAs implies that knocking out or knocking down these miRNAs would directly affect the expression patterns of key diapause-related target genes (such as Kr-h, JHE, JHEH, FOXO, Cry, and Per) and subsequently influence diapause in Ostrinia furnacalis.

In your response, you indicated that functional studies of key miRNAs should include experiments such as double luciferase reporter assays, miRNA agomir/antagomir injections to analyze gene expression patterns and phenotypic changes during diapause, and investigations into the effects of exogenous hormone treatments. This is a valid and constructive approach; however, if the author doesn't have this result in this study, I believe at least that this should be reflected in the discussion section of your manuscript, not just a reply for the author's future research directions.

Specificly, I strongly suggest incorporating a discussion on potential future directions, such as the injection of miRNA agomirs/antagomirs to analyze the expression patterns of key genes in diapause or the potential impact of exogenous hormone treatments (e.g., juvenile hormone or ecdysone) on miRNA and target gene expression patterns. Please refer to the study [https://doi.org/10.1371/journal.pgen.1008762; https://doi.org/10.1016/j.dci.2021.104036] as a relevant reference.

2. Finally, I recommend including a conceptual model diagrame or flowchart as a concluding figure in the manuscript. This diagram could illustrate the procedure to identify miRNA, how some miRNAs might influence diapause development in Ostrinia furnacalis, providing a visual summary of your findings and hypotheses for future research directions.

Comments on the Quality of English Language

In addition, I noticed some overlap between the discussion and results sections. Some results part should be included in detail, and the discussion part should not describe a lot.

Round 3

Reviewer 1 Report

Comments and Suggestions for Authors

I don't have any major concerns; the author of the manuscript has largely addressed my concerns.

The only additional suggestion is to include a discussion part on Bombyx mori exhibiting ELA behavior under the influence of BmNPV, and treat with the injection of dopamine or dopamine receptor agonist (as recommended in the second review letter for the suggested reference).

Lastly, the reference formatting in the manuscript seems inconsistent. I strongly recommend that the author carefully review and standardize the reference formatting manually, beyond relying on software like EndNote or Mendeley, as these tools can sometimes make errors.

Comments on the Quality of English Language

Please double-check the Reference format one by one.
